# N, S Co-Coordinated Zinc Single-Atom Catalysts for N-Alkylation of Aromatic Amines with Alcohols: The Role of S-Doping in the Reaction

**DOI:** 10.3390/nano13030445

**Published:** 2023-01-21

**Authors:** Xueping Zhang, Qiang Zhang, Jiacheng Reng, Yamei Lin, Yongxing Tang, Guigao Liu, Pengcheng Wang, Guo-Ping Lu

**Affiliations:** 1School of Chemistry and Chemical Engineering, Nanjing University of Science & Technology, Xiaolingwei 200, Nanjing 210094, China; 2School of Chemistry and Life Sciences, Suzhou University of Science and Technology, Suzhou 215009, China; 3School of Food Science and Pharmaceutical Engineering, Nanjing Normal University, Wenyuanstreet 200, Nanjing 210032, China

**Keywords:** zeolite imidazole frameworks, zinc single atom catalyst, N, S co-doped carbon, N-alkylamine, borrowing hydrogen

## Abstract

S-doping emerged as a promising approach to further improve the catalytic performance of carbon-based materials for organic synthesis. Herein, a facile and gram-scale strategy was developed using zeolitic imidazole frameworks (ZIFs) as a precursor for the fabrication of the ZIF-derived N, S co-doped carbon-supported zinc single-atom catalyst (CNS@Zn_1_-AA) via the pyrolysis of S-doped ZIF-8, which was modified by aniline, ammonia and thiourea and prepared by one-pot ball milling at room temperature. This catalyst, in which Zn is dispersed as the single atom, displays superior activity in N-alkylation via the hydrogen-borrowing strategy (120 °C, turnover frequency (TOF) up to 8.4 h^−1^). S-doping significantly enhanced the catalytic activity of CNS@Zn_1_-AA, as it increased the specific surface area and defects of this material and simultaneously increased the electron density of Zn sites in this catalyst. Furthermore, this catalyst had excellent stability and recyclability, and no obvious loss in activity after eight runs.

## 1. Introduction

Amines are highly valuable compounds in the field of fine chemicals and have broad applications in agrochemicals, pharmaceuticals, dyes and catalysts [1,2,3,4,5]. The N-alkylation of amines via the hydrogen-borrowing process is green and efficient for the preparation of amines owing to its step and atom economy [6,7,8,9]. In this field, noble metal catalytic systems such as Ru [10], Rh [11], Ir [12], Re [13], Ag [14], Au [15] and Pt [16], have been applied widely; meanwhile, the hydrogen-borrowing capacity of non-noble metal-based catalysts, including Fe [17,18,19], Co [20], Ni [21,22], Cu [23], Hf [24], Cr [25], Mn [26,27] and Mo [28], has also been explored (Figure 1). Although a large number of efficient catalysts have been developed for this reaction, the exploration of novel catalysts to improve the catalytic efficiency and reduce costs is still one of the keys to the development of this field.

More recently, the construction of C-C and C-N bonds via the hydrogen-borrowing pathway catalyzed by zinc has attracted the attention of researchers. In 2020, Mannathan’s group first reported on the zinc nitrate hexahydrate-catalyzed N-alkylation of amines with alcohols [29]. However, this protocol suffered from limits, including harsh conditions and an unrecyclable catalyst. Our group developed a lignin-derived Zn single-atom/N-doped porous-carbon catalyst (LCN@Zn-SAC) for the hydrogen auto-transfer α-alkylation of aromatic ketones with alcohols, in which the Zn electron density is inversely proportional to the reaction energy barriers [30]. Therefore, the coordination between the electron-rich Zn sites and the less-electronegative C, P and S [31,32] may improve the borrowing hydrogen ability of Zn sites.

Single-atom catalysts (SACs) have been widely applied in heterogeneous catalysis because of their definite metal active sites (MASs), high atomic utilization efficiency and excellent performance [33,34,35]. Their relatively simple and clear structure is an excellent model for studying the electronic and geometric structure of MASs, which provides a feasible strategy for the investigation of their catalytic mechanism [36,37,38,39]. N-doped carbon materials are cheap and powerful supports for SACs, in which nitrogen can stabilize single-atom metals effectively [40,41,42].

Zeolite imidazole frameworks (ZIFs) have excellent pore structure and dopability, and high nitrogen content, which are ideal precursors for the synthesis of N-doped carbon-supported single-atom catalysts [43,44,45,46]. Nevertheless, the preparation of ZIFs suffers from several limits, such as a slow nucleation rate and large size of ZIFs [47,48,49]. Both amines and ball-milling can accelerate the nucleation of ZIFs and reduce ZIFs’ particle sizes [35,50,51,52]. Based on the above results, we reason that ball milling in the presence of amines may be an alternative strategy for the synthesis of ZIF to accelerate nucleation and produce smaller ZIF particles.

Inspired by these reports, we were prompted to explore the possibility of heteroatom-doped Zn-SAC in N-alkylation. Herein, we report a S-doped ZIF-8 produced by one-pot ball-milling, and the ZIF-derived N, S co-doped carbon-supported zinc single-atom catalyst (CNS@Zn_1_-AA) was obtained by pyrolysis. Using thiourea as a sulfur source, ammonia and aniline were used to accelerate the precipitation of ZIFs and control the size of ZIFs [51]. This synthetic approach has several advantages, including its simple and scalable procedure, small amount of solvent, inexpensive raw materials and relatively high zinc loading (3.88 wt.%). Compared with previous non-noble metal catalysts, the performance of this catalyst was among the best in the N-alkylation of aromatic amines with alcohols (120 °C, TOF up to 8.4 h^−1^) (Figure 1).

## 2. Results and Discussion

### 2.1. Characterization of CNS@Zn_1_-AA

As illustrated in Figure 1, ZIF-8-AA@S was prepared by ball-milling, in which thiourea, aniline and ammonia were used as the additives. Then, CNS@Zn_1_-AA was obtained by the pyrolysis of ZIF-8-AA@S. Other Zn@NCs, whose naming rules are shown in Table 1, were prepared by the same procedure, except for the types of additives.

The roles of aniline and ammonia in the synthesis of ZIFs were investigated by control experiments and SEM and XRD tests. Spherical ZIFs were formed in the presence of aniline (Figure 2a,b), while irregular flake and rod ZIFs were generated in the absence of aniline (Figure 2c,d). Meanwhile, the XRD characterization proved that aniline was essential for the formation of stable ZIFs’ crystal structures (Figure 2e). The addition of ammonia did not change the crystal structure, but had a significant effect on the particle size of ZIFs. The particle size order of ZIFs was ZIF-8-An@S > ZIF-8@S > ZIF-8-Am@S > ZIF-8-AA@S. Therefore, it can be concluded that (1) aniline, as a coordination modulator, could adjust the morphology of ZIF and promote self-assembly to obtain ZIFs with a stable crystal structure [51]; (2) ammonia played a role in accelerating the nucleation of ZIFs and reducing the particle size of ZIF [53].

The role of S-doping in the material was investigated by BET and Raman tests. Both N_2_ adsorption isotherms of CN@Zn_1_-AA and CNS@Zn_1_-AA were type IV characteristic curves. The BET surface area could be improved by S-doping (964 m^2^/g vs. 1080 m^2^/g), while the average pore diameter decreased from 2.9 nm to 2.1 nm (Appendix A, Appendix A). According to the Raman spectra, the order of I_D_/I_G_ was CN@Zn_1_-AA (1.026) < CNS@Zn_1_-AA (1.048) (Appendix A), indicating that S-doping reduced the graphitization degree and promoted the formation of the pore structure and defects [54,55], which is consistent with the BET results. CNS@Zn_1_-AA presented two broad peaks at 24° and 43°, which were assigned to the (002) and (101) crystal planes of graphitic carbon materials. No Zn signal was observed, excluding the presence of large crystal particles of the Zn species (Appendix A).

The chemical state of CNS@Zn_1_-AA was investigated by XPS (Appendix A). The peaks at 284.3 eV, 284.8 eV, 285.1 eV and 286.9 eV belonged to C-S, C=C, C-N (or C-C) and C=O (Appendix A) [56]. There were four peaks, 398.3 eV (pyridinic N), 399.5 eV (Zn–N), 401.3 eV (pyrrolic N) and 403.1 eV (graphitic N), in the XPS spectrum of N 1s (Appendix A) [57]. In the S 2p spectrum (Appendix A), the binding energy peaks located at 164.1 and 165.4 eV could be assigned to the 2p_3/2_ and 2p_1/2_ spin-orbitals of the C/Zn–S–C group, and the peaks at 167.9 eV and 169.2 eV belonged to oxidized S and N-S [58]. The XPS spectrum of Zn 2p had two relatively weak peaks centered at 1021.5 eV (Zn 2p_3/2_) and 1044.9 eV (Zn 2p_1/2_) (Appendix A) [59]. In addition, the amount of Zn in CNS@Zn_1_-AA was determined by ICP-MS to be 3.88 wt.% (Appendix A).

No highly crystalline Zn species were observed via TEM (Figure 3a), which was consistent with the results of XRD. Furthermore, high-angle annular dark-field STEM (HAADF-STEM) was performed to investigate the Zn configurations in CNS@Zn_1_-AA at the atomic level (Figure 3b). Large amounts of bright dots marked with red circles represent isolated Zn atoms. The EDX element mapping images showed uniformly distributed N, O, S and Zn signals, indicating that Zn, N and S were successfully doped into the carbon matrix (Figure 3c–f).

The electronic and coordination structure of the Zn sites in CNS@Zn_1_-AA were further unraveled at the atomic level by X-ray absorption near-edge structure (XANES) and extended X-ray absorption fine structure (EXAFS) fitting [60,61,62]. As shown in Figure 4a, the Zn K-edge X-ray absorption near-edge structure (XANES) of CNS@Zn_1_-AA was compared with those of Zn foil, ZnO and zinc phthalocyanine (ZnPc). The edge positions of CNS@Zn_1_-AA were located between those of Zn foil and ZnO, demonstrating that the average oxidation state of Zn was between 0 and +2 [63,64,65]. EXAFS fitting was carried out to analyze the chemical configuration of Zn atoms in CNS@Zn_1_-AA. The peak at 1.63 Å was observed by the contribution curves of Zn-N (1.56 Å) and Zn-S (1.83 Å) (Figure 4b), and no obvious peak was located at the position of Zn-Zn coordination (2.30 Å) (Figure 4c), in agreement with the above XRD and HAADF-STEM results, which further confirmed the existence of Zn as a single atom in CNS@Zn_1_-AA. After fitting with the IFEFFIT package, the local atomic bond coordination number ratio between the Zn-N and Zn-S scattering paths was close to 3.7:1 (Appendix A). Based on the above results, the chemical coordination configuration of CNS@Zn_1_-AA could be understood as the Zn single-atom centers coordinated with 3~4 N and 1 S.

### 2.2. Catalytic Performance

The catalytic performance of CNS@Zn_1_-AA was investigated by choosing the alkylation of aniline with benzyl alcohol as a model reaction (Table 2). Neither the catalyst nor the base alone could achieve good results (entries 1 and 2). The activity of commercial zinc catalysts is poor (entries 4–6). The catalytic activity of CNS@Zn_1_-AA was significantly higher than that of CN@Zn_1_-AA due to S doping (entries 3 and 7). Aniline and ammonia were also necessary to maintain the catalytic activity of CNS@Zn_1_-AA (entries 3, 8 and 9), which can be attributed to the more stable crystal structure and smaller size of ZIFs, thereby resulting in this material having a larger specific surface area. After screening different reaction times and bases, the combination of 12 h and KOH was the best option (entries 10–13).

To further confirm the general applicability of this catalyst, the substrate scope of N-alkylation was investigated (Figure 2). Both anilines containing electron-withdrawing groups (-Br) and electron-donating groups (-Me, -Ph) (**3a**–**3e**) could be applied in the protocol. In the cases of 2-aminopyridine and 2-aminobenzothiazole, the target products (**3f**, **3g**) were obtained in a high yield. Adenine failed to react with benzyl alcohol (**3h**), and long-chain fatty amines exhibited poor reactivity in this system (**3i**, **3j**). Both 2-thiophenemethanol and 2-pyridinemethanol resulted in the production of the corresponding products in high yields (**3k**, **3l**). Inert aliphatic alcohol could also react with aniline to yield the final products (**3q**–**3u**), but a higher temperature was required in most cases. Only imine could be formed in the reaction of cinnamyl alcohol with aniline (**3o**). 4-Chloro-N-(2-furylmethyl) aniline (**3v**) and 4-bromine-N-(2-furylmethyl) aniline (**3w**) were also obtained with high yields by this protocol, both of which were crucial intermediates for the synthesis of pharmaceuticals [66,67,68].

### 2.3. Kinetic Experiments and DFT Calculations

In order to confirm the type of mechanism of the hydrogen-borrowing process (Meerwein–Ponndorf–Verley (MPV)-type or metal hydride-type), the reaction of o-hydroxychalcone with benzyl alcohol over CNS@Zn_1_-AA was performed (Scheme S1). The C=C bond of o-hydroxychalcone was reduced selectively, while the carbonyl group would be selectively reduced in the MPV reaction [69]; so, this reaction was more inclined to be a metal hydride-type reaction.

In the H/D kinetic isotope effect (KIE) experiment (Figure 3), the *k*_H_/*k*_D_ value was measured by parallel experiments using benzyl alcohol (PhCH_2_OH) or isotope-labeled benzyl alcohol (PhCD_2_OH) as substrates, obtaining a value *k*_H_/*k*_D_ of 2.65. Hence, the C–H cleavage of benzyl alcohol should be included in the rate-determining step (RDS) [19].

Furthermore, the energy barriers for the dehydrogenation of benzyl alcohol (RDS) at different zinc sites were calculated according to the density functional theory (DFT). The order of energy barriers for the dehydrogenation of benzyl alcohol at different single Zn sites was ZnN_3_S < ZnN_4_ < ZnN_4_S, which was proportional to the Zn positive-charge density (Figure 5 and Appendix A). Therefore, the coordination environment of Zn sites was adjusted by rationally doping heteroatoms to obtain more electron-rich Zn single-atom sites, which could effectively improve the hydrogen-borrowing ability of Zn-SAC [26,54].

### 2.4. Recyclability of CNS@Zn_1_-AA

Finally, the recyclability of CNS@Zn_1_-AA was investigated under the optimized conditions (Figure 6). CNS@Zn_1_-AA was recycled by centrifugation, washed with EtOAc and directly used for the next run, and only a minor loss in yield was observed after eight runs. The TEM, HAADF-STEM and XRD results suggested that no obvious metal agglomeration was detected after eight runs (Appendix A).

## 3. Conclusions

In summary, a simple and scalable strategy was developed for preparing a ZIF-derived N, S-co-doped carbon-anchored Zn single-atom catalyst (CNS@Zn_1_-AA) by the ball-milling and pyrolysis processes. S-doping played a crucial role in the activity of this material: (1) It increased the defects and specific surface area of CNS@Zn_1_-AA; (2) it enhanced the electron density of Zn sites, thereby improving the catalytic activity of CNS@Zn_1_-AA, which was confirmed by both experimental and theoretical calculation results. This material exhibited excellent performance in the hydrogen auto-transfer alkylation of aromatic amines (120 °C, TOF up to 8.4 h^−1^). To the best of our knowledge, this is the first example of the Zn-SAC-catalyzed N-alkylation of aromatic amines via the hydrogen-borrowing strategy, in which inert aliphatic alcohols could be also applied. Furthermore, this catalyst showed excellent stability, and no significant activity degradation was observed after eight runs.

## Data Availability

Not applicable.

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
