# Peer review of "N, S Co-Coordinated Zinc Single-Atom Catalysts for N-Alkylation of Aromatic Amines with Alcohols: The Role of S-Doping in the Reaction"

_nanomaterials, 2023, doi:10.3390/nano13030445_

Round 1

Reviewer 1 Report

The manuscript entitled "N, S co-coordinated zinc single-atom catalysts for N-alkylation of aromatic amines with alcohols: The role of S-doping on the reaction" presents the results for the fabrication of the zeolite imidazole frameworks-derived N, S co-doped carbon supported zinc single-atom  catalyst for the hydrogen auto-transfer alkylation of aromatic amines.

A list of shortcomings that need revision is given below:

1. Table S2  data need to be corrected. The content of Zn is 38 times larger than of the total composite: 38826 mg Zn/Kg CNS@Zn1-AA. Maybe the first unit is wrong and is also inconsistent with line 69 comments.

2. In the title, I suggest a point mark, instead of two points.

N, S co-coordinated zinc single-atom catalysts for N-alkylation 2 of aromatic amines with alcohols. The role of S-doping on the reaction

3. ZIF abbreviation has to be explained before its first introduction; zeolite imidazole frameworks (ZIF) 

4. The turnover frequency (TOF) is also unexplained.

5. To a better understanding/ accuracy, please preserv the same order of ZIFs in Table 1 and in Figure 2.

6. Please complete the text:

Therefore, it can be concluded that (1) aniline, as a coordination  modulator, can XXXXXX the morphology of ZIF and promote self-assembly to obtain ZIF with stable crystal structure [51];

Reviewer 2 Report

The current paper (communication) reports the role of doping on catalyst towards chemical reactions. In general, the paper is well structured and well writing. It needs to address few minor issues as outlined below. Based on that, I recommend minor revision of the manuscript. The specific comments are as follows:

1.      Line 16: Better to introduce full form, before start using abbreviation (e.g. ZIF).

2.      Line 20: The degree sign is not the proper one, please revise and like this in the rest of the paper.

3.      The last paragraph of the introduction section should be the ‘aim’ of the present work, which is missing.

4.       Fig. 1 needs ref.

5.      Fig. 3, please include indexed SAD pattern of the TEM micrograph.

6.      Fig. 6: Include error bars, if possible/applicable.

7.      Line 213: It should be expressed in third person!

8.      In acknowledge section, lines 232-234 should be avoided!
